# Scoring of Growth of Unruptured Intracranial Aneurysms

**DOI:** 10.3390/jcm9103339

**Published:** 2020-10-18

**Authors:** Seppo Juvela

**Affiliations:** Department of Clinical Neurosciences, University of Helsinki, FI-00029 Helsinki, Finland; seppo.juvela@helsinki.fi; Tel.: +358-50-5457258

**Keywords:** unruptured intracranial aneurysm, natural history, cigarette smoking, risk factors, aneurysm growth, subarachnoid hemorrhage

## Abstract

The purpose was to obtain a reliable scoring for growth of unruptured intracranial aneurysms (UIAs) in a long-term follow-up study from variables known at baseline and to compare it with the ELAPSS (Earlier subarachnoid hemorrhage, Location of the aneurysm, Age > 60 years, Population, Size of the aneurysm, and *S*hape of the aneurysm) score obtained from an individual-based meta-analysis. The series consists of 87 patients with 111 UIAs and 1669 person-years of follow-up between aneurysm size measurements (median follow-up time per patient 21.7, range 1.2 to 51.0 years). These were initially diagnosed between 1956 and 1978, when UIAs were not treated in our country. ELAPSS scores at baseline did not differ between those with and those without aneurysm growth. The area under the curve (AUC) for the receiver operating curve (ROC) of the ELAPSS score for predicting long-term growth was fail (0.474, 95% CI 0.345–0.603), and the optimal cut-off point was obtained at ≥7 vs. <7 points for sensitivity (0.829) and specificity (0.217). In the present series UIA growth was best predicted by female sex (4 points), smoking at baseline (3 points), and age <40 years (2 points). The AUC for the ROC of the new scoring was fair (0.662, 95% CI 0.546–0.779), which was significantly better than that of ELAPSS score (*p* < 0.05). The optimal cut-off point was obtained at ≥4 vs. <4 points for sensitivity (0.971) and specificity (0.304). A new simple scoring consisting of only female sex, cigarette smoking and age <40 years predicted growth of an intracranial aneurysm in long-term follow-up, significantly better than the ELAPSS score.

## 1. Introduction

Subarachnoid hemorrhage (SAH) is a serious disease with a high rate of unfavorable outcomes [1,2,3,4]. The rate of treatment of unruptured aneurysms (UIAs) has been increased with a purpose to reduce incidence of SAH [4,5,6]. However, the effect of treating UIAs on the incidence SAH has been low [7], compared with that of reducing the prevalence of smoking [7,8]. In any case, the majority of diagnosed UIAs never rupture during the remainder of the patient’s lifetime even in the case of patients of working age [9,10].

Since the prevalence of aneurysms is not decreasing [11], the risk of rupture is likely decreasing, resulting in lower SAH incidence rates [7,8]. The indications for treatment of UIAs are already challenging because of a lack of prospective studies of natural history of UIAs [12]. Since such studies are not possible to complete nowadays without treatment selection, interest has been focused on studies of aneurysm growth, assuming this to be an important prognostic factor for aneurysm rupture [13,14,15]. Most UIA growth studies have also been derived from patient populations with treatment selection bias with low rupture rates. UIA growth has been shown to increase risk of aneurysm rupture [14,15].

According to two recent meta-analyses of aneurysm growth, female sex, cigarette smoking at baseline, aneurysm size, posterior circulation aneurysms, and possibly age, hypertension, and aneurysm multiplicity may increase the risk of UIA growth, while prior SAH and a family history of aneurysms do not [13,14]. Aneurysm growth rates were lower in prospective and high-quality studies [13].

A large meta-analysis of individual data from 1507 patients in 10 cohorts showed UIA growth in 257 cases (17%) during 5782 patient-years of follow-up [16]. Predictors for aneurysm growth during a mean follow-up of 3.8 years were *E*arlier SAH (decreased risk), *L*ocation of the aneurysm, *A*ge >60 years, *P*opulation, and *S*ze and *S*hape of the aneurysm (ELAPSS score). The 3-year growth rate ranged between <5% and >42% and the 5-year growth rate between <9% and >60%, depending on the risk factors. The results between ELAPSS score and previous meta-analyses differed between each other for sex, age, previous SAH, populations, and some aneurysm locations. Cigarette smoking was not studied in the ELAPSS study. Most of the cohorts were from selected populations where the patients with UIAs were those left for conservative follow-up because of an estimated low rupture risk, or a limited life expectancy or high treatment risks [16]. UIA may, however, be unchanged >20 years, then grow a little and rupture [15,17,18].

An almost lifelong follow-up cohort of UIAs [9,15] seen at our institution was not subject to treatment selection bias and had been considered to be the highest quality study and to have the least source of bias of any prospective natural history studies of both UIA rupture and growth [12,13,14]. The aim here is to investigate the validity of the ELAPSS score in predicting the long-term UIA growth risk and to search for a new score from variables known at baseline which could better predict the UIA growth rate.

## 2. Methods

### 2.1. Patient Population

Approval for the surveys and follow-up data collection was obtained from the local ethics committee (the ethics committee of the Department of Neurosurgery, Helsinki University Central Hospital, Helsinki, Finland; the record 6/94; 29 December 1994) and all aspects of the study were in compliance with national legislation and the Declaration of Helsinki [9]. All the patients gave their written informed consent.

The series comprised of 87 patients with UIAs diagnosed at the Department of Neurosurgery of Helsinki University Central Hospital between 1956 and 1978, when UIAs were not operated on in Finland and when this hospital was responsible for neurosurgical services for almost the whole Finnish population [1,9,10,15,17,18]. The series included patients with angiographic follow-up and also those whose aneurysms were measured at autopsy, since aneurysm size is known be associated with both aneurysm rupture and case fatality after SAH [1,15,17].

### 2.2. Age and Sex Distributions of Patients with and without Follow-Up Monitoring

Of the 142 patients with UIAs diagnosed before 1979, 56 patients were studied in the outpatient department during 1996–1998 [15,17]. Only 4 patients with UIAs who were alive and had not been examined previously with control angiography failed to visit the outpatient department for CT angiography [17]. These patients were elderly (>80 years) and/or had a severe incapacitating disease and were accordingly excluded from the study.

In addition, the aneurysm sizes of 43 patients were monitored with conventional or digital angiography (*n* = 33) or at autopsy (*n* = 10) between 1959 and 2008 because of an aneurysm rupture or a need to check the aneurysm status. The last follow-up was performed in 2012.

Since the cohort included patients who were admitted during a time period of 22 years (1956–1978), there were a further 51 patients with UIAs who had already died without a control angiography, vast majority of them from unrelated causes [15,17,18]. As was to be expected, these patients who were excluded without a control examination of their UIA (55 out of 142) differed from those in the present cohort by their higher age at baseline (mean ± SD, 47.1 ± 9.2 vs. 38.4 ± 9.1 years, *p* < 0.001).

### 2.3. Follow-Up Methods

Detailed follow-up protocols have been reported previously [1,9,10,15,17,18,19,20,21]. Briefly, the follow-up evaluations with reports were based on postal questionnaires and telephone interviews obtained from patients or proxies every 10 years from the 1960s onwards. In total, 56 patients were additionally studied in the outpatient department during 1996–1998 [15,17]. I personally interviewed the patients with the use of a structured questionnaire, and their cerebral vessels were studied with 3D CT angiography (CTA). The structured questionnaire through the follow-up included patient characteristics, previous diseases, hospital visits, medication, and health behavior.

Additional information on all the patients was obtained from medical records supplied by other hospitals and general practitioners, and the accuracy of the medical data including blood pressure (BP) values was recorded [1,9,10,15,17,19]. Autopsy reports and official death certificates were examined for all deceased patients. In Finland, a statutory medico-legal autopsy is performed on all those who die due to trauma or unknown causes (Act on Inquests into the Cause of Death, 459/1973, Finnish Law). The follow-up was complete.

### 2.4. Risk Factors

Core and most highly recommended supplementary variables for the study of UIA were available for the present purpose [22], and the ELAPSS score [16] was recorded for each patient from variables at the baseline. Hypertension at baseline was defined as a systolic pressure repeatedly >140 mm Hg, a diastolic pressure >90 mm Hg, or use of antihypertensive medication. Cigarette smoking was grouped as follows: never a smoker, formerly a regular cigarette smoker (quit before or during the follow-up), or currently a cigarette smoker at the end of follow-up [9,15,17,18,19]. Alcohol consumption was calculated as approximate grams of absolute ethanol consumed within one week (1 standard drink = 12 g of alcohol) throughout the follow-up period. A family history of SAH was defined as ≥2 first-degree relatives with verified ruptured aneurysms.

### 2.5. Aneurysm Measurements

All angiographies and CTAs were examined by 2 experienced neuroradiologists who had no knowledge of the patients’ case histories [1,9,15,17,18,19,20,21]. Discrepancies were resolved by consensus opinion. The maximum diameters and locations of the UIAs were measured from standard projections of 2D conventional angiograms at baseline; also recorded were the location and shape (round, oval, irregular, or lobulate) of the UIA. Aneurysm growth rates were measured as the difference in the maximum aneurysm diameter between the initial and follow-up angiographies from the same projections or the initial angiography and the autopsy [15,17,18].

### 2.6. Statistical Analysis

The data were analyzed with IBM SPSS Statistics, version 25.0, for Windows (IBM Corp., Armonk, NY, USA). The ELAPSS score variables at baseline were compared according to the UIA’s growth status using Fisher’s exact test or the Pearson Chi-square test. Sum scores (expressed as mean ± standard deviation (SD) and median with interquartile range (IQR)) were compared by means of *t*-tests and the Mann–Whitney *U*-test.

Odds ratios (ORs) with 95% confidence intervals (CIs) of the risk factors for growth of UIA were analyzed by multivariable unconditional logistic regression. A maximum-likelihood stepwise forward elimination procedure was used, with selection of the variables to be added into the model on the basis of the magnitudes of their probability values (*p* < 0.15). The new integer risk score points for aneurysm growth were based on the regression coefficients in the final multivariable logistic regression model. The areas under the curve (AUC) for the receiver operating curves (ROC) were calculated for both the ELAPSS and the new growth scores. The C-statistic with 95% CIs was used to test how the total scores matched the observed rupture rates, the AUC between the scores being analyzed using the technique of Hanley and McNeil [23]. A two-tailed *p* value < 0.05 was considered statistically significant.

### 2.7. Data Availability

Anonymized data not published within this article are available by request from any qualified investigator.

## 3. Results

### 3.1. Patient Characteristics and Follow-Up

A total follow-up time of 87 patients was 1669 years between aneurysm measurements, and the mean follow-up time for aneurysm size per patient was 19.2 years (median 21.7, range 1.2 to 51.0 years). Twenty-seven patients (31%) suffered from a SAH and of them, 10 were fatal. The mean measurement follow-up time in these 27 patients was 12.0 years (range 1.2 to 24.1 years), and that for the 60 patients without a rupture was 22.4 years (1.3 to 51.0 years). All 27 UIAs which later ruptured had increased in size by at least 1 mm and out of the largest UIAs of 60 patients without an aneurysm rupture, 13 increased by ≥1 mm.

### 3.2. ELAPSS Score and Aneurysm Growth

Factors of ELAPSS score stratified by aneurysm growth ≥1 mm are shown in Table 1. None of separate factors differed by growth of UIA. The total score points were also similar between growth groups. Since all patients were Finns exclusion of population from the ELAPSS score (7 points) had no effect on significance levels. Odds ratio of ELAPSS score for aneurysm growth was 0.97 per unit (95% CI, 0.90–1.04, *p* = 0.35).

### 3.3. Risk factors for UIA Growth and the New Growth Score

All the ELAPSS factors with additional UIA locations, sex, hypertension, smoking, alcohol consumption, and family history were tested with logistic regression analysis to identify the variables which best predicted a future aneurysm growth. Women (adjusted OR 3.6, 95% CI 1.2–10.7, *p* = 0.019), smoking at baseline (2.7, 0.93–7.6, *p* = 0.068), age <40 years (2.1, 0.8–5.5, *p* = 0.15) predicted best long-term UIA growth. Smoking at the end of follow-up associated better with growth (3.2, 1.1–8.9, *p* = 0.029) than smoking at baseline but it cannot be used in a new score since it is not a predictive variable but represents better a long-term habit (Table 2). Total score sum was higher in those with a UIA growth (*p* = 0.003). A new system for scoring of UIA growth was constructed from the regression coefficients of the three variables: women 4 points, smoking 3 points, and age <40 years 2 points (Table 2). Odds ratio of new scoring for aneurysm growth was 1.4 per unit (95% CI, 1.1–1.8, *p* = 0.006).

### 3.4. The AUC for the ROCs of UIA Growth Scores

The AUC for the ROC of the new scoring for aneurysm growth was fair (0.662, SE 0.060, 95% CI 0.546–0.779, *p* = 0.013), and the optimal cut-off point was obtained at ≥4 vs. <4 points for sensitivity (0.971) and specificity (0.304) (Figure 1). The AUC for the ROC of simple scoring (points 0 vs. 1 for all three variables) was almost the same (AUC 0.658, SE 0.060, 95% CI 0.540–0.776, *p* < 0.015). The optimal cut-off point was obtained with the sum score at ≥2 vs. <2 points with a sensitivity (0.771) and specificity (0.457). The AUC of the ELAPSS score was fail (0.474, SE 0.066, 95% CI 0.345–0.603, *p* = 0.69), and the optimal cut-off point was obtained at ≥5 vs. <5 points for sensitivity (0.829) and specificity (0.217) (Figure 1). So, false positive rate was high. The difference between the scores for AUCs was significant (*p* < 0.05). The AUC of ELAPSS, which also included patients with missing values for smoking was 0.450 (SE 0.063, 95% CI 0.327–0.573, *p* = 0.42) and the cut-off point at ≥5 vs. <5 points yielded sensitivity 0.800 and specificity 0.213 (Table 3).

## 4. Discussion

The growth score of UIAs obtained from this prospective study with an almost lifelong follow-up and a very low treatment selection bias suggests that growth of UIAs among patients of working age can be estimated quite reliably with only three variables at baseline. Female sex, patient age (<40 years), and cigarette smoking determine the risk of UIA growth significantly better than does the ELAPSS score.

Simultaneously with the discovery of increasing numbers of UIAs, it is likely that the indications for treatment will decrease, because of the decline in the prevalence of smoking and the incidence of SAH [7,8] and because of the discovery of incidental UIAs in older people more than previously. These patients are also less likely to be smokers or to have higher treatment risks because of age. The cost-effectiveness ratio of UIA treatment is thus increasing. To obtain more reliability in the treatment decision, follow-up angiographic investigations have become more prevalent to search for UIAs that grow. Growing of UIAs correlates with later rupture, particularly in a long-term follow-up [14,15,17,18].

According to the results of two recent meta-analyses of aneurysm growth, female sex, cigarette smoking at baseline, aneurysm size, posterior circulation aneurysms, and possibly age, hypertension, and aneurysm multiplicity may increase the risk of UIA growth, while prior SAH and a family history of aneurysms do not [13,14]. There was a substantial heterogeneity for risk factors between studies, except in the case of female sex, smoking at baseline, and hypertension. High data quality (low risk of study bias), Japanese and Finnish populations, and prospective study design were associated with decreased growth risk [13]. The results of these meta-analyses were subject to treatment selection of UIAs and follow-ups were mostly short extending only approximately for 3–5 years. Because of treatment selections most UIA growth studies have shown low rupture rates [13,14]. High risk UIAs (large UIAs in younger patients and smokers) are usually treated already at baseline without any follow-up [12].

The present study with an almost lifelong follow-up of UIA patients also included patients with a high UIA rupture risk as was seen in the relatively high UIA growth rate among patients of working age. Expectedly, UIA growth score obtained from the present score included partly same factors as treatment score based on risk factors of UIA rupture [21]. The treatment score (the AUC for the ROC 0.755) predicted slightly better aneurysm rupture than did present growth score for the growth of UIA. Both these scores included cigarette smoking, patient age <40 years, and female gender, the latter being of borderline statistical significance in the treatment score. Aneurysm size and some locations were included in the treatment score but not in the growth score. This is not surprising since e.g., aneurysms in the anterior communicating artery are more prone to rupture with a smaller size than those in middle cerebral artery which grow more before rupture [15].

Previously published results from this cohort [15,17,18] have been reported to have the best data quality and the lowest bias of all UIA growth studies to date, in spite of reporting the lowest approximate annual incidence of growth [13,14]. ELAPSS score study [16] with a relatively short follow-up (median 2.5 years) has some contradictory results with PHASES score study which was planned to predict aneurysm rupture risk (e.g., for previous SAH and some aneurysm locations) [12]. Although ELAPSS score study is a pooled individual patient data meta-analysis its results also conflict with those of two other recent meta-analyses for sex, age, previous SAH, populations, and some aneurysm locations [13,14]. Predictors for aneurysm growth were *E*arlier subarachnoid hemorrhage (decrease risk!), *L*ocation of the aneurysm, Age > 60 years, *P*opulation, *S*ize of the aneurysm, and *S*hape of the aneurysm (ELAPSS). Of the total 1507 patients with UIAs only 18 patients (1.2%) had an aneurysm rupture, and of 257 growing UIAs only 8 (3.1%) ruptured. Cigarette smoking which has been most significant risk factor for UIA growth and SAH [24,25] was not analyzed in ELAPSS score study. UIA growth in long-term follow-up also correlated better with its rupture [14,15]. Mean follow-up among patients of working age between diagnosis and rupture in the present study was >10 years. Routine follow-up angiography during 1–3 years after the diagnosis seems also to be unreliable since aneurysm may be stable >20 years, then grow 1 mm and rupture [15,18].

Selection criteria for aneurysm occlusion treatment and conservative follow-up with/without control angiograms in participating centers in the ELAPSS score study were heterogeneous. In some countries, small UIAs in elderly are routinely followed radiologically. In Finland only those patients with uncertainty for treatment are followed up on clinical-based criteria. This was seen also in the ELAPSS score study where patients with angiographic follow-up in Finland were significantly younger than elsewhere and these patients are known to be more likely smokers and have higher UIA growth risk [13,15,17,18]. This has leaded arbitrarily to a high ELAPSS score among Finns although in one meta-analysis, the growth rate of their UIAs was even lower [13]. ELAPSS score describes preferably the growth of low rupture risk UIAs in patients left to conservative treatment. This was also seen in the low aneurysm rupture rate in the ELAPSS score study. These results are difficult to be generalized to all patients with UIAs.

The strengths of this study lie in the complete and almost lifelong follow-up of patient of working age and the very limited treatment selection bias [9,15,18]. Correspondingly, a previous aneurysm growth study based on this cohort [15,17,18] was considered to be of high study quality and have low sources of bias relative to other UIA growth studies [13,14]. Although Finnish people have been considered to be subject to a higher risk of aneurysm rupture, the incidence of SAH is no higher in Finland than elsewhere when standardized for the study design with its inclusion and exclusion criteria, the accuracy of diagnosis, and the sex and age distributions of the population [8]. The Nordic countries seem to have similar incidences of SAH. A recent twin study has shown that the contribution of genetic factors as the cause of SAH appears to be slight, and the main causal factor seems to be cigarette smoking [26].

A limitation of this study is the relatively small sample size, despite a long total follow-up time as compared with other large prospective study populations, which on the other hand, had shorter follow-ups and a high treatment selection bias. Patients with a prior history of SAH have not been shown to have a significantly higher risk of aneurysm rupture or growth than others when confounding factors are taken into account [12]. In this study, all initial UIA measurements were obtained from two-dimensional conventional angiograms, which may yield less accurate measures than three-dimensional angiograms. For UIA growth measurement the same projections were done as was done in initial angiograms.

The present simple, rapid scoring system for UIA growth is significantly better and easier to use than the ELAPSS score. This new growth score contains also the same factors than the treatment score obtained from this cohort. The latter score is directed to evaluate rupture risk of UIAs and its value for prediction is somewhat better than that for growth score. The treatment score is preferable in clinical practice for treatment decision and it likely reduces need of follow-up angiographies.

## Figures and Tables

**Figure 1 jcm-09-03339-f001:**
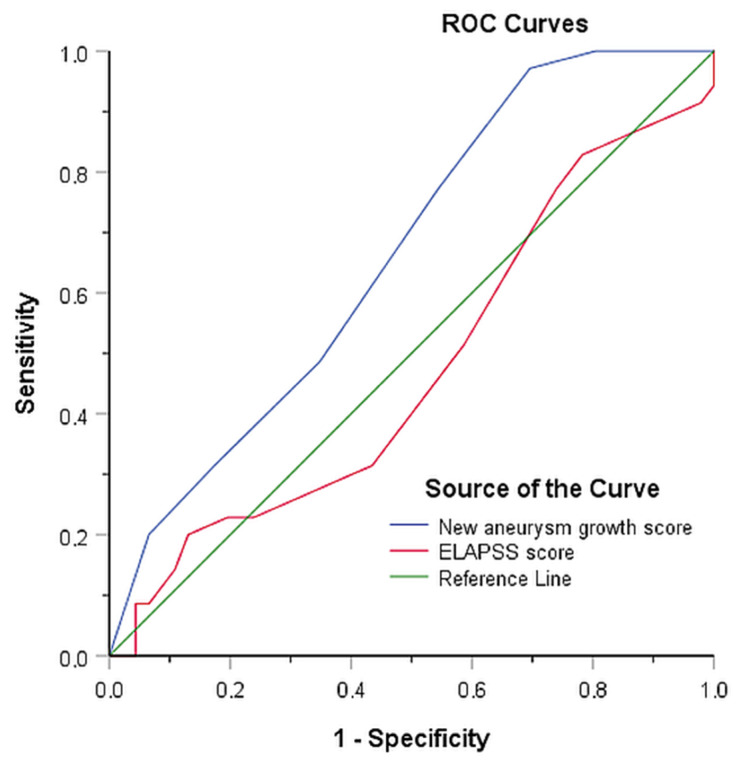
Receiver operating characteristic curves of growth scores for predicting the long-term growth of unruptured intracranial aneurysms. See text for statistics of the curves. The difference between the scores for area under the curve (AUC) was significant (*p* < 0.05). ROC = Receiver operating characteristic.

**Table 1 jcm-09-03339-t001:** ELAPSS score stratified by long-term aneurysm growth.

Factor	ELAPSS Score Points	Patients with Aneurysm Growth (*n* = 40, 46%)	Patients wo Aneurysm Growth (*n* = 47, 54%)	All Patients (*n* = 87, 100%)
**(E) Earlier SAH from another aneurysm (%)**				
Yes	0	37 (47)	42 (53)	79 (100)
No	1	3 (38)	5 (63)	8 (100)
**(L) Location of the aneurysm (%)**				
ICA/ACA/ACOM	0	9 (38)	15 (63)	24 (100)
MCA	3	21 (49)	22 (51)	43 (100)
PCOM/posterior	5	10 (50)	10 (50)	20 (100)
**(A) Age (%)**				
< 60 years	0	40 (46)	47 (54)	87 (100)
>60 years (per 5 years)	1	0	0	0
**(P) Population (%)**				
Finland	7	40 (46)	47 (54)	87 (100)
**(S) Size of the aneurysm (%), mm**				
1.0–2.9	0	8 (73)	3 (27)	11 (100)
3.0–4.9	4	17 (45)	21 (55)	38 (100)
5.0–6.9	10	8 (32)	17 (68)	25 (100)
7.0–9.9	13	5 (63)	3 (37)	8 (100)
≥10.0	22	2 (40)	3 (60)	5 (100)
**(S) Shape of the aneurysm**				
Regular	0	37 (49)	39 (51)	76 (100)
Irregular	4	3 (27)	8 (73)	11 (100)
**Mean ELAPSS score by aneurysm growth (SD)**		16.2 ± 6.0	17.9 ± 6.5	17.3 ± 6.2
**Median ELAPSS score (IQR)**		15 (12–20)	17 (12–21)	16 (12–20)

ELAPSS = Earlier subarachnoid hemorrhage, Location of the aneurysm, Age > 60 years, Population, Size of the aneurysm, and *S*hape of the aneurysm; SAH = subarachnoid hemorrhage; ACOM = anterior communicating artery; ACA = anterior cerebral artery; ICA = internal carotid artery; SD = standard deviation; IQR = interquartile range (range between the 25th and 75th percentiles); MCA = middle cerebral artery; PCOM = posterior communicating artery.

**Table 2 jcm-09-03339-t002:** New growth score stratified by long-term aneurysm growth.

Factor	Aneurysm Growth Score Points	Patients with Aneurysm Growth (*n* = 40, 46%)	Patients without Aneurysm Growth (*n* = 47, 54%)	All Patients (*n* = 87, 100%)
**Sex**				
Men	0	11 (32)	23 (68)	34 (100)
Women	4	29 (55)	24 (45)	53 (100)
**Current cigarette smoking at baseline**				
No	0	13 (35)	24 (65)	37 (100)
Yes	3	22 (50)	22 (50)	44 (100)
**Current cigarette smoking at end of follow-up**				
No	N/A	16 (34)	31 (66)	47 (100)
Yes	N/A	19 (56)	15 (44)	34 (100)
**Age**				
<40 years	2	25 (52)	23 (48)	48 (100)
≥40 years	0	15 (38)	24 (62)	39 (100)
**Mean score by aneurysm growth (SD)**		5.9 (1.8) *	4.4 (2.4)	5.1 (2.3)
**Median score (IQR)**		5 (5–7) *	5 (3–6)	5 (4–6)

IQR = interquartile range (range between the 25th and 75th percentiles). Current smoking value was missing for 1 of the 47 patients without an aneurysm growth (2%) and 5 of the 40 with growth (13%). * *p* = 0.003 (*p* = 0.011 with Mann–Whitney *U*-test), those aneurysm growths had a higher growth score; *n* = 81, those with missing smoking habit were excluded from comparison.

**Table 3 jcm-09-03339-t003:** Aneurysm growth scores and number of patients with an aneurysm growth.

Aneurysm Growth Risk Score Points	Patients with Aneurysm Growth (%)	Patients without Aneurysm Growth (%)	No. of Patients (%)
**ELAPSS score ***			
0–4 (7–11)	8 (44)	10 (56)	18 (100)
5–7 (12–14)	12 (57)	9 (43)	21 (100)
8–11 (15–18)	7 (50)	7 (50)	14 (100)
12–16 (19–23)	8 (35)	15 (65)	23 (100)
17–30 (24–37)	5 (45)	6 (55)	11 (100)
Total	40 (46)	47 (54)	87 (100)
**Present score** ^†^			
0–3	1 (7)	14 (93)	15 (100)
4	7 (50)	7 (50)	14 (100)
5–6	16 (48)	17 (52)	33 (100)
7–9	11 (58)	8 (42)	19 (100)
Total	35 (43)	46 (57)	81 (100)

* ELAPSS score groups are shown both with (in parentheses) and without Finnish population points (7 points) which has no effect on significance levels. The difference between ELAPSS scores and aneurysm growth were not significant. Cut-off point at ≥5 vs. <5 points yielded sensitivity 0.800 and specificity 0.213. ^†^
*p* < 0.01, for differences between aneurysm growth and scoring groups. Use of ≥4 vs. <4 points in the present score for aneurysm growth gives a sensitivity of 0.971 and specificity of 0.304.

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
