# Peer review of "Scoring of Growth of Unruptured Intracranial Aneurysms"

_jcm, 2020, doi:10.3390/jcm9103339_

Round 1

Reviewer 1 Report

The author presented a series of 87 patients with 111 UIAs, aiming to obtain a reliable scoring for the growth of unruptured intracranial aneurysms. The study is appropriately designed, and the results are presented clearly and in detail. The discussion is precise and interesting. No major grammar or spelling mistakes were identified. No plagiarism was detected either. Prognosis and epidemiology of unruptured intracranial aneurysms is a topic of interest among neurosurgeons, many other specialties .

Author Response

I thank the reviewer for his/her highly complimentary and encouraging comments of this paper. He/she checked carefully the manuscript and recommended no revisions.

Reviewer 2 Report

Excellent work, of great interest to all colleagues who care for intracranial aneurysm patients. Knowledge about factors that influence risk of rupture has grown remarkably within the past decade, and especially growth is highly alarming - but can only be observed in follow up imaging and might be noticed too late. Scoring the risk of growth is important.

I like the idea of keeping it simple. This is not a not simple piece of work at all, it is work of high statisitical complexity - but the result is easier to use than similar scores: it predicts the risk of growth with less criteria, and even better.

Hopefully, the decision to treat or not will never been taken just by calculating numbers. In UIAs, though, it is always a weigh up of estimatet risks, and for estimating a good score is very helpful.

Two typos found:

line 96 - through, not thorough

line 242 - earlier, not Earlier

Author Response

I thank the reviewer for his/her highly complimentary and encouraging comments of this paper. He checked carefully the manuscript and recommended only correction of two typos.

- Thorough is now replaced by through, which is correct word.

- Earlier subarachnoid hemorrhage with a capital letter E is part of ELAPSS score. In the sentence “Predictors for aneurysm growth were Earlier subarachnoid hemorrhage (decrease risk!), Location of the aneurysm, Age >60 years, Population, Size of the aneurysm, and Shape of the aneurysm (ELAPSS).” All score letters are shown as capital letters and thus do not need revision.

Reviewer 3 Report

The author has investigated the predictive scoring of unruptured intracranial
aneurysms (UIAs) and compared with the ELAPSS score on the growth risk. From the references the author has recently published a very similar study using the same cohort follow-up subjects to show cigarette smoking is the predictive risk factor for UIA growth and rupture, with similar conclusion and study design, which decreases the novelty of the current manuscript. Moreover, a study published with larger sample size demonstrated ELAPSS is a reliable score for prediction of UIA growth(J Stroke. 2019 Sep;21(3):340-346. 
External Validation of the ELAPSS Score for Prediction of Unruptured Intracranial Aneurysm Growth Risk), which the author should cite and discuss in the manuscript. Suggestion for the author to expand further or add to the sample size in order to increase the strength and novelty of the study.

Author Response

I thank the reviewer for his/her encouraging comments of this paper. As shown in the paper the study was based on a long-term follow-up with a true natural history until SAH, death or year 2012. This is the only natural history study since in the other studies the patients with UIAs with elevated rupture risk were treated already at baseline without any follow-up or follow-up of few months. This causes high selection bias and the results are difficult to be generalized to all UIAs.

This cohort has been source of several reports during elapsed five decades. The reviewer refers to the study shown in the reference list with the number 15 (Juvela, S. Growth and rupture of unruptured intracranial aneurysms. J Neurosurg 2019, 131, 843–851). This study dealt with risk factors for growth of UIAs leading to rupture. This was a pure risk factor study of a subpopulation of growing UIAs. The designs of the present growth score study are different. The purpose here was to obtain a scoring from significant or almost significant risk factors. The growth score included also other variables than cigarette smoking which was the solitary risk factor of UIA growth leading to rupture. This scoring could help practicing clinicians to estimate simply by scoring which UIAs likely will grow in the future, even during the remaining lifetime. Scoring is a much easier method than is to estimate it simultaneously from odds ratios of several risk factors. This is also seen in the comments of reviewers 1 and 2.

The reviewer wanted previous validation study of ELAPSS score to be cited in this paper (J Stroke. 2019 Sep;21(3):340-346). Unfortunately, the previous study had similar concerns with patient selection leading to study bias as had ELAPSS score study. Patients with UIAs with elevated rupture risks were excluded by treatment selection already at baseline. This was seen in both validation and development cohorts where rupture rates of UIAs were low, also among growing UIAs. C-statistic of cohorts for aneurysm growth were relative low (approximately 0.7) already in short-term follow-up (until 3 and 5 years after the diagnosis). Most important shortage was removing of female sex and cigarette smoking from scores which have been shown to be the most important risk factors for aneurysm growth and rupture. So, it is not reasonable to expand discussion by the study published in Journal of Stroke. Discussion would be only repetitious and does not specify the message of the present study.

As is shown in the 2nd paragraph of the Introduction, expansion of the sample size to increase study power is nowadays impossible for UIAs without treatment selection bias. Expansion would be possible only to include patients with UIAs with low rupture risk into the cohort. Combining of patients without treatment selection bias with those with a selection bias leads to results which are also biased. Unfortunately, there are not nowadays available new patient cohorts which include patients whose aneurysms have similar rupture risks than those in a population where UIAs are not treated and have true natural history.

Reviewer 4 Report

An interesting and significant paper. Some minor grammar corrections required (e.g. line 139 "will be made available"- is available).

Statistical question for Author. On the bottom of the Table 2 Median score (IQR) of patients with aneurysm growth is 5, same 5 is for patients without aneurysm growth. The difference (?) is marked as statistically significant. Either it is a mistyping or requires an explanation.

Author Response

I thank the reviewer for his/her highly complimentary and encouraging comments of this paper. He/she checked carefully the manuscript and recommended few minor revisions.

- Available wording is now corrected. Previous phrase was used in earlier papers.

- In Table 2, numbers have been checked several times. Mean scores 5.9 vs. 4.4 differed significantly when parametric test (t-test) was used (p =0.003). This was correct. I wanted also to show nonparametric comparison with Mann Whitney U- test. Although medians of the groups are the same (5), the distributions using IQRs were different (5-7 vs. 3-6). Mann Whitney U- test showed p- value 0.011 which is significant although more conservative that the significance level obtained by t-test.

Round 2

Reviewer 3 Report

The author has carefully answered the review's concerns and revised accordingly, no further comment for publication.